# Controlled Synthesis of Chromium-Oxide-Based Protective Layers on Pt: Influence of Layer Thickness on Selectivity

Myles Worsley [1,2], Vera Smulders [1] and Bastian Mei [1,*]

1   PhotoCatalytic Synthesis Group, Faculty of Science and Technology, MESA+ Institute for Nanotechnology, University of Twente, Meander 229, P.O. Box 217, 7500 AE Enschede, The Netherlands
2   College of Engineering, Design and Physical Sciences, Brunel University London, Middlesex, London UB8 3PH, UK
*   Correspondence: b.t.mei@utwente.nl

**Abstract:** Chromium-oxyhydroxide ($Cr_xO_yH_z$)-based thin films have previously been shown in photocatalysis and industrial chlorate production to prevent unwanted reduction reactions to occur, thereby enhancing the selectivity for hydrogen evolution and thus the overall process efficiency. Here, a highly reproducible synthesis protocol was developed to allow for the electrodeposition of $Cr_xO_yH_z$-based thin films with controlled thickness in the range of the sub-monolayer up to (>4) multilayer coverage. Electrodeposited $Cr_xO_yH_z$ coatings were electrochemically characterized using voltammetry and stripping experiments, allowing thickness-dependent film selectivity to be deduced in detail. The results are discussed in terms of mass transport properties and structure of the electrodeposited chromium oxyhydroxide films. It is shown that the permeation of diatomic probe molecules, such as $O_2$ and CO, was significantly reduced by films as thin as four monolayers. Importantly, it is shown that the prepared thin film coatings enabled prolonged hydrogen oxidation in the presence of CO (up to 5 vol.%), demonstrating the benefits of thin-film-protected electrocatalysts. In general, this study provides insight into the synthesis and use of thin-film-protected electrodes leading to improvements in (electro)catalyst selectivity and durability.

**Keywords:** ultrathin oxide layers; electrocatalysis; hydrogen oxidation; selectivity; electrodeposition; eQCM

## 1. Introduction

The generation of hydrogen from renewable sources is of great interest for application in the sustainable production of chemicals. Among others, photocatalytic water splitting is often seen as a promising technology that solely uses semiconductor particles and solar illumination [1–4]. To achieve an acceptable rate of $H_2$ production, a semiconductor with a band gap of approximately 1.8 V is required [5–7]. Diminishing charge recombination rates and improving the kinetics of the surface reactions require the use of nanoparticulate electrocatalysts (also referred to as co-catalysts) on the semiconductor surface [8–10], where noble metal catalysts, such as Pt and Rh are considered to be the most effective for hydrogen evolution (HER). Unfortunately, noble metal co-catalysts are also known to catalyse competing reactions, such as the oxygen reduction reaction (ORR), thereby decreasing the overall efficiency of the process [11–13].

Enhanced selectivity towards HER can be achieved through the use of core-shell structures. These structures are used to reduce or prevent undesirable side reactions from occurring. Examples of such protective layers include $NiO_x$ [6,14,15], Mo-based coatings [16–18], and oxides or oxyhydroxides of Si, Ce, Ti, Nb, and Ta [19–22]. Of particular interest is chromium oxide ($CrO_x$), which has long been used in the chlorate industry to improve the overall process efficiency [23–27]. Similarly, $CrO_x$-based thin films or shells on noble metal catalysts have also been shown to improve the efficiency of photocatalytic water splitting [28,29] where, in particular, protected Rh particles are used, although in

general, uncoated Pt nanoparticles are more active for hydrogen evolution [30]. The effect of thin film coatings has been validated by various studies and is widely accepted [31–34]. Nevertheless, little attention has been paid to the controlled, synthesis-enabling high control of the thickness of the coatings. Generally, thinner layers are preferred as there are limitations on the use of $CrO_x$, particularly in higher oxidation states, falling under Annex XIV of the REACH legislation [35].

$CrO_x$ coatings are presumed to enable selective hydrogen evolution by functioning as a permeable membrane for $H^+$ and $OH^-$ rather than an electroactive surface themselves [36]. In addition, it has been suggested that with the Grotthuss mechanism, i.e., in thin hydrated films, protons reach the catalyst surface via proton hopping, enabling hydrogen evolution but preventing various larger molecular species from permeating [37]. As such, Qureshi et al. showed that a wide variety of molecular species that are readily reducible on uncoated noble metal electrode surfaces are prevented from undergoing reactions when chromium oxyhydroxide protective layers are present [38]. In addition to the protection against molecular species, the protection of common contaminants, such as CO, is of great relevance, i.e., for hydrogen fuel cell and compression applications [39], and as such, controlled synthesis of over-coatings and a detailed understanding of the functionalities of thin-film-coated electrocatalysts is required.

Here, by using carefully controlled (sub- to multi-)monolayer electrodeposition, we assessed the HER activity along with the material's ability to suppress the oxygen reduction (ORR) back reaction in alkaline media. For the different coating thicknesses prepared, the functionality and stability of the $CrO_x$ thin film coatings under operational conditions were determined. The coatings were further characterised using $H_2$ and CO as probe molecules to elaborate on the permeability of small dissolved molecules. By the careful preparation of $Cr_xO_yH_z$ thin films it was revealed that a minimum thickness of four monolayers of $Cr_xO_yH_z$ is required for the complete suppression of the $O_2$ back reaction, whereas a single monolayer is ineffective at significantly reducing activity in any of the reactions tested. Most importantly, it is finally shown that the physical openness of the polynuclear Cr oxyhydroxide structure in acidic electrolyte enables prolonged activity for hydrogen oxidation even in the presence of CO (5 vol.%).

## 2. Results

To enable electrodeposition of $Cr_xO_yH_z$ thin films on Pt electrodes with a high degree of thickness control, a facile protocol was developed. Using cyclic voltammetry (CV) and electrochemical quartz crystal microbalance (eQCM) measurements in different $Na_2Cr_2O_7$ concentrations, deposition and dissolution of $Cr_xO_yH_z$ films were assessed allowing for precise thickness determination. In cyclic voltammetry, reduction of the dissolved $Cr^{VI}O_4^{2-}$ anions occurs with a broad peak at $E_{max} = 0.15$ V vs. RHE and the resulting $Cr_xO_yH_z$ film is oxidised above 1 V vs. RHE in the anodic sweep (data are shown in Figure S1 in the Supplementary Materials), being comparable with previous studies [36–38]. As expected for both reduction and oxidation peaks, the charge ($Q$) increases with concentration and in the case of the electrochemical response due to oxidation, the peak potential shifts more positively with the amount of material deposited. To reliably quantify the amount of deposited material, the charge associated with the oxidative response was compared to the mass change ($\Delta m$) determined using eQCM measurements (Figure S2), allowing for the definition of the relationship between the charge required to remove the $Cr_xO_yH_z$ coating and the mass deposited on the Pt electrode. Approximate thicknesses ($\tau$) of the coatings were estimated from the mass change based on the composition and known densities of chromium oxyhydroxides [40], considering either fully hydrated $Cr(OH)_3$ ($\tau_{max}$) or dehydrated and deprotonated $Cr_2O_3$ ($\tau_{min}$) forms of chromium oxide.

To prepare coatings of different thicknesses for further electrochemical analysis, electrodeposition was undertaken by chronoamperometry (CA) at 0.15 V vs. RHE for 10 min using $Na_2Cr_2O_7$ concentrations ranging from 1 μM to 1 mM at logarithmically increasing intervals. The charge determined by oxidative stripping of these coatings using linear sweep

voltammetry (LSV) allowed for thickness determination following the eQCM-derived charge–mass relationship. Table 1 summarizes the concentration dependence of $\tau$ for both hydrated and dehydrated forms of the thin film. As thin films prepared by electrodeposition have been proposed to be a mixture of both compositions, for simplicity here, the layer thickness was considered to be an average between the two compositions and is referred to in the following as $\tau_{av}$ [25,27,41–43]. At the lowest concentration (1 µM), a $\tau_{av}$ value of only 0.14 nm was determined to be below previous estimates of $0.5 \pm 0.2$ nm for a single monolayer coverage of $Cr(OH)_3$ [44], indicating incomplete (sub-monolayer) coverage. For the coating to be useful, a homogeneous layer across the entire electrode surface is required, which, according to the developed strategy is achieved at a $Na_2Cr_2O_7$ concentration of 10 µM, where the derived thickness ranges well around the literature's estimate of a single monolayer. At a concentration of 100 µM, a growth of 2–4 monolayers was estimated and a further increase to the concentrations of 1000 µM resulted in an estimated coverage of 4 monolayers. As summarized in Table 1, the developed protocol allowed for reproducible synthesis of $Cr_xO_yH_z$ on Pt electrode. In comparison with earlier work, coatings prepared at 100 µM (2–4 monolayers) are comparable with the thinnest film tested (1.8 nm) [36]. Finally, it is worth noting that the coatings are homogenous as suggested by SEM analysis (Figure 1).

**Table 1.** Estimated minimum (dehydrated) and maximum (fully hydrated) $\tau$ after 10 min CA at different concentrations of $Na_2Cr_2O_7$.

| Conc. (µM) | $\tau_{min}$ (nm) | $\tau_{max}$ (nm) | No. of Monolayers |
|---|---|---|---|
| 1 | $0.10 \pm 0.02$ | $0.17 \pm 0.03$ | <1 monolayer |
| 10 | $0.46 \pm 0.09$ | $0.77 \pm 0.14$ | 1 monolayer, approx. |
| 100 | $1.26 \pm 0.07$ | $2.11 \pm 0.12$ | 2–4 monolayers |
| 1000 | $1.76 \pm 0.04$ | $2.95 \pm 0.07$ | 4+ monolayers |

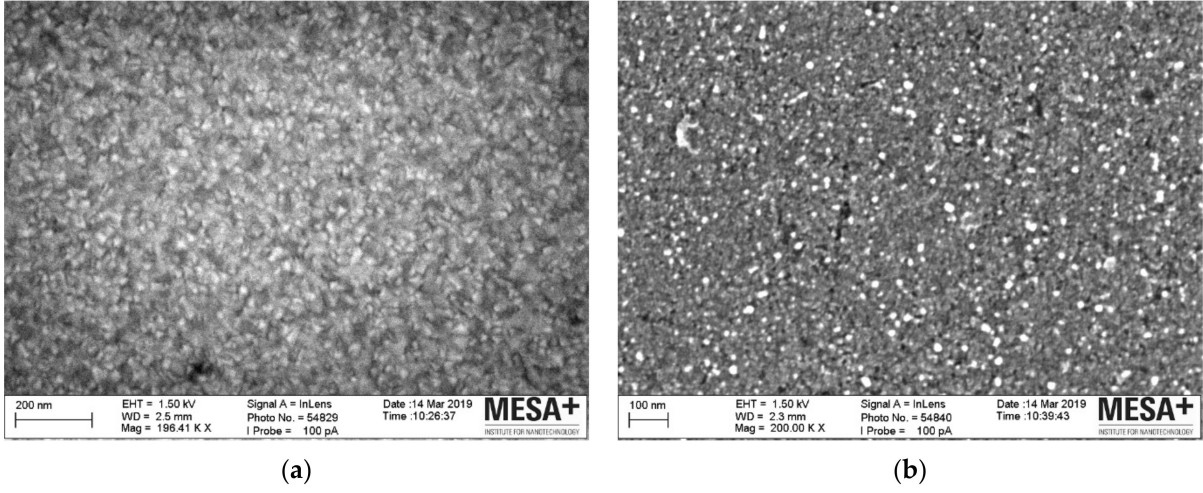

(**a**)          (**b**)

**Figure 1.** SEM micrographs of the (**a**) uncoated Pt surface and (**b**) $Cr_xO_yH_z$-coated Pt surface produced from electrolyte containing 5 mM $Na_2Cr_2O_7$/0.1M NaOH.

Linear sweep voltammetry was used to study hydrogen evolution on $Cr_xO_yH_z$-modified Pt electrodes. The presence of multiple layers does not block the production of $H_2$ at the Pt electrode, as can be seen in Figure 2, and has been previously reported. Still the overpotential increases with coverage. With around a monolayer or less of $Cr_xO_yH_z$ coverage on Pt the increase in overpotential is rather negligible (about 5 mV), whereas beyond 2 monolayers a significant cathodic shift by approximately 25–30 mV is observed. For

the thickest layers (4+ layers) 35–40 mV larger overpotentials are required when compared to uncoated Pt. This is also confirmed by Tafel analysis using:

$$\eta = a + b \log j$$

where *b* is the Tafel slope. Table 2 provides a summary of the determined exchange current densities ($j_0$) and Tafel slopes (*b*) for hydrogen evolution at different thicknesses of $Cr_xO_yH_z$ on Pt, being in agreement with previous literature [45].

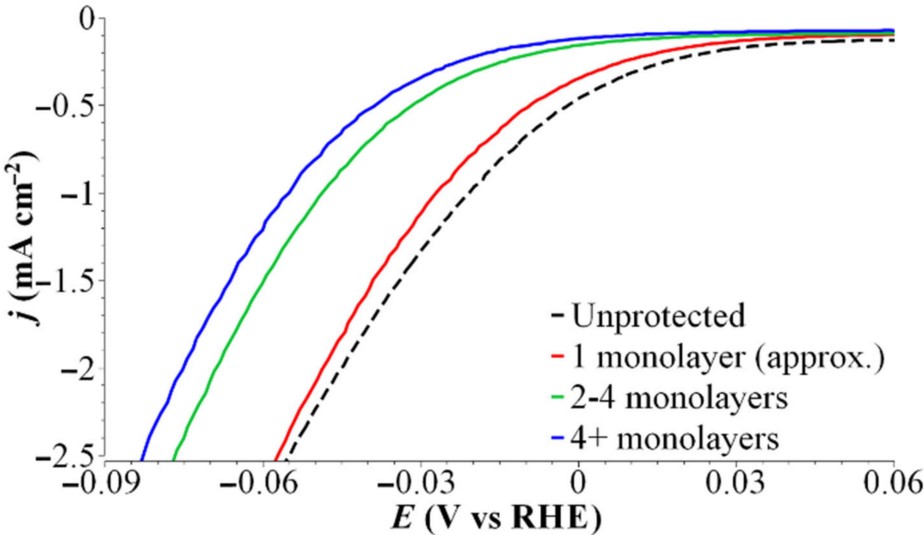

**Figure 2.** Linear sweep voltammograms (LSVs) in the HER regime with different levels of $Cr_xO_yH_z$ coverage in $N_2$-saturated 0.1 M NaOH (aq) at 50 mV s$^{-1}$, at 900 rpm.

**Table 2.** Exchange current densities ($j_0$) and Tafel slopes (*b*) for HER with different thicknesses of $Cr_xO_yH_z$ on Pt.

| $\tau_{av}$ (nm) | $j_0$ (A cm$^{-2}$) | *b* (mV dec$^{-1}$) |
|---|---|---|
| 0 (Pt only) | −3.33 | 64 |
| 0.62 (1 monolayer, approx.) | −3.45 | 60 |
| 1.69 (2–4 monolayers) | −3.85 | 57 |
| 2.36 (4+ monolayers) | −4.00 | 55 |

With the successive growth of $Cr_xO_yH_z$, the current for hydrogen evolution drops, particularly from 1 to 2+ layers. If the reaction proceeded via a tunnelling mechanism the current density would be expected to decrease exponentially with increasing coverage [37,46], whereas in a mechanism proceeding via proton transfer, it would be expected to be invariant [36]. Thus, the obtained results suggest tunnelling to be the dominant mechanism for electrodes covered with a few layers of $Cr_xO_yH_z$, and with increasing thickness, a proton transfer mechanism appears to be dominating. Further decreases are attributable to changes in the concentration gradient across the layer as it gets thicker, reducing the overall proton concentration at the Pt surface [47].

The polarisation curves for measurements in the presence of gas phase reactants ($H_2$ and $O_2$) under identical RDE mass transport controlled conditions are shown in Figure 3. The presence of one monolayer slightly lowers the measured limiting current density of the hydrogen oxidation reaction (HOR) and the oxygen reduction reaction (ORR) compared to bare Pt. In addition, higher potentials are required in the mixed-controlled regime to obtain similar current densities. Additional layers reduced the limiting current density significantly in both cases, which indicate that at >2, the coating becomes increasingly impermeable. The partial blocking of the ORR using $CrO_x$-modified Pt electrodes with 2–4 monolayers was even maintained during extended operation (see Figure S3 in the

Supplementary Materials). Nevertheless, even for samples with >4 monolayers, the activity for $H_2$ oxidation was not entirely lost, whereas ORR did not occur in the potential range probed, i.e., $O_2$ was effectively blocked by the $Cr_xO_yH_z$ coating. This supports earlier reports indicating that ORR is strongly supressed by the presence of a $CrO_x$-based coatings whilst allowing the permeation of protons and subsequent hydrogen evolution to occur [36].

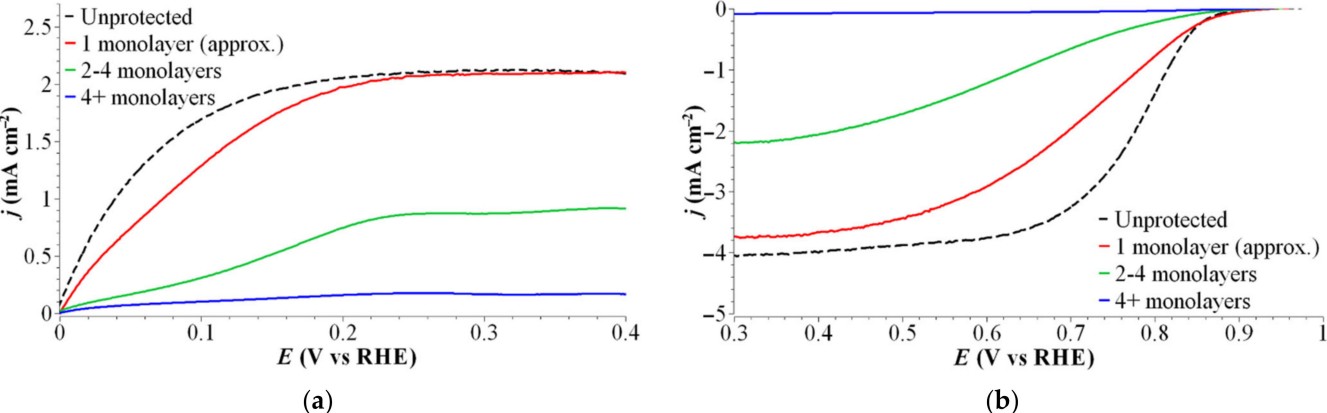

**Figure 3.** Linear sweep voltammograms (LSVs) of the (**a**) HOR ($H_2$ saturated) and (**b**) ORR ($O_2$ saturated) with different levels of $Cr_xO_yH_z$ coverage, both in 0.1 M NaOH (aq) at 50 mV s$^{-1}$, at 900 rpm.

The $H_2$ and $O_2$ permeability of overlayers was further quantified using the measured limiting current densities [48–51]. Compared to the bare Pt electrode—where mass transport properties, i.e., the diffusion coefficient $D$, can be obtained by Levich analysis (see Figures S4–S6)—transport for $Cr_xO_yH_z$-coated Pt must be associated with both transport in overlayers and through the diffusion boundary layer. Levich analysis correlates the limiting current density ($j_l$) and rotation rate ($\omega$) according to:

$$j_{l,Pt} = 0.62 \frac{nFD^{2/3}c}{v^{1/6}} \omega^{1/2}$$

where $n$ is the number of electrons transferred, $F$ is the Faraday constant (96,485.33 C mol$^{-1}$), $c$ is the bulk electrolyte concentration given by the solubility of the reactant, and $v$ is the kinematic viscosity [5]. In the presence of a $Cr_xO_yH_z$ over-coating diffusion, coefficients or film permeability are obtained by the following relation describing the mass transfer resistance related to the limiting current densities associated with double layer diffusion and overlayer coating, as shown below:

$$\frac{1}{j_{l,total}} = \frac{1}{j_{l,Pt}} + \frac{\tau_{film}}{nFP_{film}c}$$

where the gas permeability $P_{film}$ is linked to the solubilities ($S_{film}$) of $H_2$ or $O_2$ in the film [52] and their film diffusion coefficient [48,51], respectively.

$$P_{film} = D_{film}S_{film}$$

Generally, the permeability of the film decreases with increasing thickness, pointing towards a change in film structure for thicker coatings. Using the approximated over-coating thicknesses, film permeabilities for hydrogen and oxygen at the highest film thickness, i.e., at >4 monolayers of $2.7 \pm 0.7 \times 10^{-10}$ cm$^2$ s$^{-1}$ and of $5.2 \pm 1.3 \times 10^{-11}$ cm$^2$ s$^{-1}$, are obtained. As already indicated by the remaining hydrogen oxidation activity observed in LSV scans permeability for hydrogen is higher compared to oxygen permeability, which is likely related to the size of the oxygen molecule.

Further performance evaluation of the $CrO_x$-coated Pt electrodes was undertaken using CO poisoning and subsequent stripping experiments. Figure 4a shows the baseline corrected stripping measurements of the bare and coated Pt electrodes (for additional information, see Figure S7). The peak position of the CO stripping peak is known to vary considerably with electrode surface and electrolyte conditions; however, of interest is the shape and area of the stripping peak, being indicative of the type(s) of surface interaction and the amount of CO adsorption [53]. It is generally accepted that the oxidation follows the Langmuir-Hinshelwood mechanism and that the chemisorbed layer is comprised predominantly of linear and bridge-bonded CO [54]. In the case of the bare Pt, a clearly defined pre-peak as well as the main stripping signal are observed. The pre-peak is related to changes in the surface CO adlayers, changing from higher density adlayers to more relaxed ones, where weakly bound bridge-bonded CO is desorbed [55–57]. Additionally, structural rearrangements during desorption include movement from weaker bound terrace sites to stronger step sites. The loss of peak multiplicity in the presence of a $Cr_xO_yH_z$ coating from 1 to 4 monolayers indicates that there is a restriction of the type of adsorption and surface rearrangements occur, suggesting a purely linearly bonded configuration is favoured for coated electrodes.

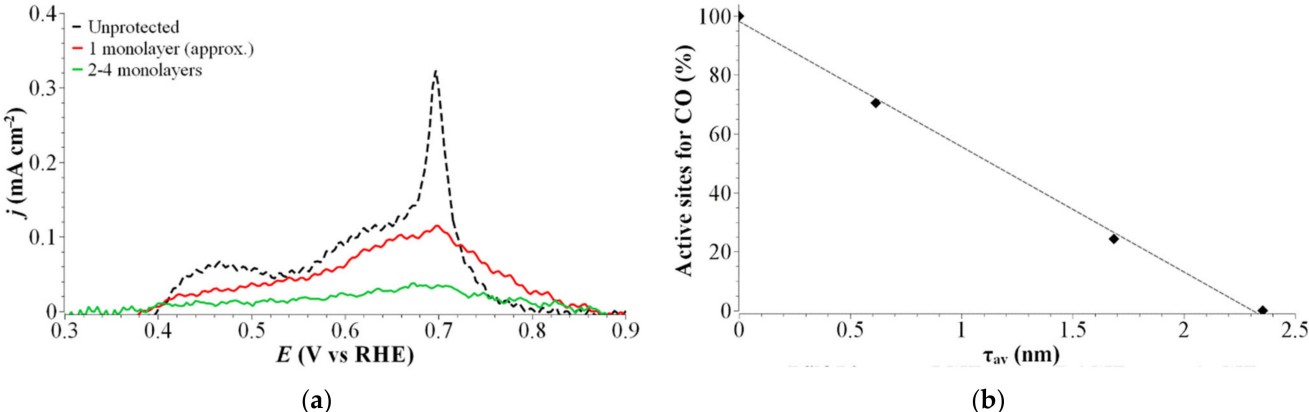

**Figure 4.** The (**a**) CO stripping peaks obtained from the CVs after poisoning for 15 min at 20% $CO/N_2$, E = 0.05 V vs. RHE, followed by $N_2$ flush for another 15 min at the same potential and (**b**) loss of active sites determined from oxidation of adsorbed CO, with linear fit.

Figure 4b shows the determined number of active sites as a function of coating thickness, for which above the layers of $Cr_xO_yH_z$, interaction of CO with the electrode is fully suppressed. The number of active sites was determined by the ratio of the charge per $cm^2$, i.e., the integrated CO stripping peak of coated and unprotected Pt surfaces. Considering this ratio at one monolayer, approximately 25% of the active sites are lost. In fact, it has previously been suggested that the presence of a $Cr_xO_yH_z$ layer on the surface of Pt reduces the number of active sites for H adsorption by around this amount [37], indicating it is still permeable to CO with just restrictions in bonding orientation. The permeability changes significantly from 1 monolayer to 2+ monolayers, an effect observed for all the reactions studied here (HER, HOR, and ORR). Thus, the thickness-controlled formation of protective $Cr_xO_yH_z$ layers on the Pt electrode suggest that a minimum of four layers (approximately 2 nm) is required to fully suppress the $O_2$ back reaction, being that it is important in photocatalytic hydrogen production. Such layers will also function as effective protective coatings and will prevent Pt electrode poisoning by CO. Whilst coatings below four monolayers do not offer complete protection against unwanted side-reaction, the efficiency towards hydrogen evolution in the presence of competing reactions will still be improved and thus the thickness-control obtained in this study provides a trade-off between activity and the amount of $Cr_xO_yH_z$ required.

## 3. Discussion

To understand the changes in protective layer effectiveness as the coating thickness increases, it is necessary to consider the structure of the coating. Polynuclear $Cr_xO_yH_z$ thin films have been studied previously and as shown in Scheme 1, are understood to be a network of hydrated Cr octahedra connected via hydrogen bonding between the OH ligands and water molecules within the complex [58–60].

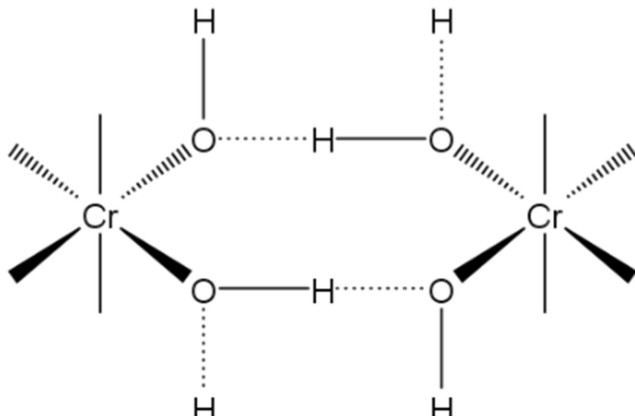

**Scheme 1.** Schematic representation of a network of hydrated Cr octahedra connected via hydrogen bonding.

A 2009 study by Torapava et al. using large angle X-ray scattering (LAXS) determined that, under alkaline conditions, Cr–O bond length was 200 pm (in the first coordination sphere) and, more importantly, Cr–Cr distances were 298 pm in polynuclear Cr(III) hydroxo complexes [61]. Up to a single monolayer, plenty of physical space for the permeation of small molecules such as $H_2$, $O_2$, or CO exists and could in part explain why activity and diffusion characteristics are not significantly different from bare Pt. Of the three, CO activity was the most reduced by the presence of a single monolayer, followed by ORR, and then HOR. Considering that $H_2$ has the smallest van der Waals radius of the three probe molecules used here, the suggested physical openness of the $Cr_xO_yH_z$ thin film is in agreement with the reported experimental observations. In addition, other factors, such as electronegativity and polarity, likely contribute to the lower permeability of the other two molecules.

The growth of thin films beyond one monolayer coverage represents a transition from a 2D to a 3D network of Cr complexes and is known to particularly favour tetramer structures in a high pH environment [62], thus making the network more restrictive to permeation. This is supported by the results obtained here, where reactions for the larger molecules $O_2$, CO, and $H_2$ to a slightly lesser extent are significantly affected as the coating grows beyond two monolayers. Further increases in the coating also lead to the possibility of partial deprotonation, particularly in the lower layers. This would result in additional closing of the structure, leading to greater protection of the Pt surface from molecular interactions.

The low permeability is thus partially caused by the electrolyte composition. To finally reveal the importance of the local pH environment, additional tests were performed in acidic electrolyte. Specifically, a coating of 4+ monolayers was prepared and was shown to greatly inhibit both CO interaction with the Pt surface and, to a large extent, to prevent oxidation of hydrogen. The Pt-coated electrode was used for the hydrogen oxidation reaction using an electrolyte saturated with 5 vol.% $CO/H_2$. With the coated Pt electrode, hydrogen oxidation occurred at a similar current density (as shown in Figure 5) and despite the significant CO concentration used here, the duration of the HOR was notably extended compared to the reference measurement using unmodified Pt surfaces. Interestingly, CO poisoning did not even result in a complete deactivation and HOR still occurred, yet at lower rates. Clearly, penetration of the CO through the coating has been slowed by diffusion

through the film. Moreover, the retention of some hydrogen oxidation activity suggests the coating is not being oxidised and sufficiently stable despite the low electrolyte pH.

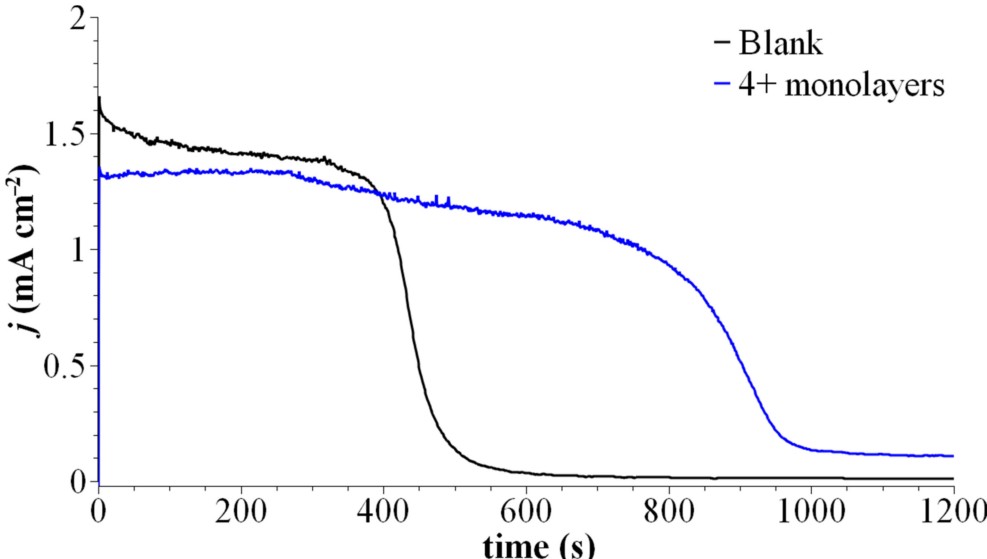

**Figure 5.** Chronoamperometry measurements at 0.2 V vs. RHE, at 900 rpm in $H_2$-saturated $H_2SO_4$ (adjusted to pH 1.8) electrolyte for blank Pt and the 4+ monolayer $Cr_xO_yH_z$-coated Pt disc, where saturating gas composition was changed to 5% $CO/H_2$ at 300 s.

Thus, apart from the direct implications in photocatalysis, this work provides additional insights into the benefits of thin-film-coated electrode surfaces. Particularly, the obtained results highlight the important factors in the preparation and development of protective layers in enhancing the selectivity of electrocatalysts toward specific reactions. The thin film structure is an important aspect of the design of such materials, as a more open structure allows for permeation of small molecules. Additionally, the physical and chemical properties of the layers directly influence the type of molecule that can easily permeate the coating or membrane as well as their behaviour to changing electrode potentials. Considering the results presented here, in future developments of thin-film-coated electrodes, multi-membrane configurations should be considered, in particular, for materials such as $Cr_xO_yH_z$ that are susceptible to oxidation but are very effective as protective layers. Further development of thin-film-coated electrodes will likely result in interesting novel ultrathin protective layers with beneficial levels of selectivity. Clearly, such developments are of importance for the implementation of emerging processes in the area of sustainable chemical production and renewable energy utilization.

## 4. Materials and Methods

Electrochemical experiments were carried out in 0.1 M NaOH electrolyte solutions. For electrodeposition of the $CrO_x$-containing protective layers, $Na_2Cr_2O_7$ (Sigma-Aldrich, St. Louis, MO, USA, >99.8%) was added to the electrolyte in concentrations ranging from 1 μM–5 mM. For deposition and subsequent characterisation, the electrolyte was pre-saturated with $N_2$ to remove air by continuous bubbling at 40 cm$^3$ min$^{-1}$ for at least 15 min. The electrochemical measurements were performed using a BioLogic VSP-10 and a 100 cm$^3$ capacity three-electrode cell consisting of a Hg/HgO ($E^0$ = 0.098 vs. RHE) or Ag/AgCl ($E^0$ = 0.197 vs. RHE) reference electrode, a Pt wire or graphite rod counter electrode, and a rotating ring disk working electrode (RRDE, Pine Research Wavevortex) composed of polycrystalline Pt (disk and ring).

The reactions of interest were studied by continuous bubbling at 40 cm$^3$ min$^{-1}$ of either $N_2$ for the hydrogen evolution reaction (HER) or $H_2$ for the hydrogen oxidation reaction (HOR) using different rotation rates. Additionally, the oxygen reduction reaction

was studied by flowing $O_2$ at the same flow rate. By adding up to 20% CO to the $N_2$ or $H_2$ stream we further evaluated the poisoning resistance of coating-modified Pt electrodes.

The mass of the deposited $Cr_xO_yH_z$ coatings was determined using a Gamry eQCM 10 M at 5 MHz with Pt coated crystals from Gamry. The difference in mass ($\Delta m$) was determined by the shift in the resonance frequency $f_s$ of the electrode using the Sauerbrey equation [63]:

$$\Delta f_s = \frac{2f_0^2 mn}{\mu_q \rho_q}$$

where $f_0$ is the fundamental frequency of the crystal, $m$ is the mass added, $n$ is the harmonic number, $\mu_q$ is the shear modulus, and $\rho_q$ is the density. This can be simplified by reducing the known constants to a calibration constant, $C_f = 56.6 \text{Hz cm}^2 \text{ g}^{-1}$, to give:

$$\Delta f = C_f m$$

where $\Delta f$ is the overall frequency change determined experimentally. All potentials are reported vs. RHE unless otherwise indicated.

## 5. Conclusions

Controlled deposition of $Cr_xO_yH_z$ protective layers on Pt has allowed for the precise definition of the required film thicknesses to allow for full suppression of $O_2$ back reaction, with CO poisoning being of relevance among others for the design of photocatalytic materials. Specifically, it is shown that a single monolayer has a minor influence on the properties of Pt electrodes and a minimum of four monolayers (approximately 2 nm) is required to fully block $O_2$ and CO. The properties of the coating are discussed on the basis of physical porosity altered by changes in the structure of the coating. Importantly, it is shown that the acidity of the electrolyte largely influences the blocking capabilities of the coating. Coatings considered to be nonpermeable in alkaline solution were shown to be active for hydrogen oxidation with extended duration in CO-containing environments. This has implications in coating design where adjustment of the structure could potentially lead to control over selectivity and in turn result in higher process efficiencies.

**Supplementary Materials:** The following supporting information can be downloaded at: https://www.mdpi.com/article/10.3390/catal12101077/s1, Figure S1: Cyclic voltammograms on Pt RRDE; Figure S2: The eQCM $\Delta m$ in 0.1 M NaOH for different $Na_2Cr_2O_7$ concentrations; Figure S3: Chronopotentiometry (CP) at $-0.2$ mA ($-1.02$ mA cm$^{-2}$) and 900 rpm in $O_2$-saturated 0.1 M NaOH (aq) for 3 different levels of coating; Figure S4: HOR linear sweep voltammograms (LSVs) at different rotation rates; Figure S5: ORR linear sweep voltammograms (LSVs) at different rotation rates; Figure S6: Levich plots; Figure S7: CO stripping sweeps; additional experimental information.

**Author Contributions:** Conceptualization, M.W. and B.M.; methodology, M.W. and V.S.; validation, M.W.; formal analysis, M.W. and V.S.; investigation, M.W. and V.S.; writing—original draft preparation, M.W. and B.M.; writing—review and editing, B.M.; visualization, M.W.; supervision, B.M.; project administration, B.M.; funding acquisition, B.M. All authors have read and agreed to the published version of the manuscript.

**Funding:** This research was funded by The Dutch Research Council (*NWO*), grant number 739.017.015.

**Data Availability Statement:** The data presented in this study are available in Supplementary Material.

**Conflicts of Interest:** The authors declare no conflict of interest. The funders had no role in the design of the study; in the collection, analyses, or interpretation of data; in the writing of the manuscript; or in the decision to publish the results.

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
