# Peer review of "Controlled Synthesis of Chromium-Oxide-Based Protective Layers on Pt: Influence of Layer Thickness on Selectivity"

_catalysts, doi:10.3390/catal12101077_

Round 1

Reviewer 1 Report

This paper reported the controlled deposition of CrOH protective layer on Pt, which allows for suppression of oxygen back reaction and CO poisoning. The results were solid and interesting. 

However, there are only electrochemical testing in the paper. It would be interesting to get some material characterization results, such as SEM to see the morphology of the film, XPS to see the composition of the film. It would make the paper more comprehensive to include more discussions about the film properties and whether or not those will affect its performance.

Reviewer 2 Report

Title: Controlled synthesis of chromium oxide based protective layers on Pt: Influence of layer thickness on selectivity

Manuscript ID catalysts-1878404

Comments:

The article reports the rational fabrication of chromium oxyhydroxide (CrxOyHz)-based thin film with a controlled thickness (~13 thickness) as protective layers on Pt by the electrodeposition method. The as-obtained materials were used for the oxygen reduction reaction, hydrogen evolution, and hydrogen oxidation in addition to CO-stripping.

The article is in line with the journal scope and could be accepted after addressing the following comments

·       The abstract and introduction should be rewritten to emphasize the novelty of this manuscript  and some results for the electrochemical performance of the materials should be mentioned in the abstract

·       The authors should explain how they calculated the active sites

·       The majority of the references are too old, so more recent references should be added especially those related to Pt-based catalysts. More references should be added like https://doi.org/10.1016/j.jelechem.2015.10.035;https://doi.org/10.1021/acssuschemeng.8b02015; 

·       The authors should add SEM results to confirm the morphologies of thus obtained thin-films

·       The figure resolutions should be improved

·       A detailed comparison table for the HER, HOR, and ORR should be added

·       In the ORR polarization curves, the diffusion-limited current density should be close to 6 mA/cm2, the authors should explain why it is lower

·       There is bulk or surface analysis for the materials like XRD or XPS

·       The CO-stripping is mainly important in the alcohol oxidation reactions because CO-poisoning Pt surface is common during the alcohol oxidation reactions, so it's useless in ORR, HOR, and HER

·       Some important references about ORR, and HER could be added like ‘’ https://doi.org/10.1016/j.elecom.2022.107207;
